# Capillary Thinning of Viscoelastic Threads of Unentangled Polymer Solutions

**DOI:** 10.3390/polym14204420

**Published:** 2022-10-19

**Authors:** Alexander Semenov, Irina Nyrkova

**Affiliations:** Institut Charles Sadron, CNRS-UPR 22, University of Strasbourg, CEDEX 2, 67034 Strasbourg, France

**Keywords:** capillary flows, polymers, viscoelasticity

## Abstract

In this paper, we theoretically consider the process of the capillary thinning of a polymer fluid thread bridging two large immobile droplets in the regime of highly stretched polymer chains. We first derive a new relation between the pressure *p* and the flow velocity *v* in unentangled polymer solutions, which is called the anti-Bernoulli law: it shows that *p* is higher where *v* is faster. Using this equation, it is shown that the flow field is asymptotically irrotational, in particular, in the thread/droplet transition zones (in the case, the negligible solvent viscosity and inertial effects). On this basis, we predict the free surface profile and the thread thinning law for the FENE-P model of polymer dynamics. The predictions are compared with recent theoretical results and some experimental data on capillary thinning.

## 1. Introduction

Long liquid cylinders tend to break in many droplets due to capillary surface forces. For a Newtonian liquid, this phenomenon, related to the well-known Plateau–Rayleigh instability, typically results in a localized pinching, leading to the formation of necks whose thicknesses rapidly decrease over time. In contrast, the thinning of polymer solution threads often result in a transient formation of nearly uniform (perfectly cylindrical) threads. The characteristic thinning time of such threads strongly increases with polymer molecular weight. This phenomenon was observed in numerous experiments [1,2,3,4], in particular those using the liquid filament rheometer (LFR) [5]. Most experiments show that the diameter of a polymer thread is thinning according to an exponential law [1,6,7].

Theoretically, such uniform exponential thinning of polymer filaments was deduced from rheological (force balance and constitutive) equations for the Oldroyd-B and FENE-P models [1,7,8]. There was, however, a problem with the early theories: the boundary condition at the filament ends was stated based on a plausible argument rather than a rigorous proof. (‘We assume that the axial stress vanished because, in the LFR, the filament is attached to large stagnant drops on stationary end plates’, as written in ref. [8].) Thus, Entov and Hinch [8] proposed that the axial polymer stress $\sigma_{p}$ in the thread of radius $a_{0}$ must be equal to the capillary pressure, $\sigma_{p}=\gamma /a_{0}$ (here, γ is the surface tension) to obtain the total axial pressure equal to the atmospheric pressure well inside the large drops at the thread ends. A similar equation, but with the prefactor 1/2 in the rhs, was obtained earlier [9] based on the slender-body approximation, which, however, is not applicable in the decisive interfacial zone between the thread and an end-droplet. Later, Clasen et al. [7] derived a different condition:(1)$$\sigma_{p}=2\gamma /a_{0}$$
applying the slender-body approximation in a different way (see also [10]). Quite recently, the above equation was re-derived [11] based on the Onsager variational principle (in the regime of negligible viscous stress due to the solvent). Equation (Equation 1) was finally rigorously established [12] for the capillary collapse of a neo-Hookean elastic cylinder in the limit of small elastic modulus *G*. It was argued [12,13] that the neo-Hookean elastic model becomes equivalent to the Oldroyd-B model in the limit of infinite polymer relaxation time, τ→∞. A further discussion of this subject is delegated to Section 4.

In the present paper, we apply the force balance equations for the Oldroyd-B model to rigorously show that in the decisive transition region near the interface between the thread and a droplet (referred to also as the entrance zone or the ‘corner flow’ zone [7]), the flow velocity $\underline{v}$, and the pressure *p* are connected by a conservation equation, which we called the anti-Bernoulli law since it shows that a faster velocity corresponds to a higher pressure. On this basis, we derive the ‘boundary condition’, confirming that Equation (Equation 1) is correct. We also show that in the regime of strongly stretched chains, the flow velocity field $\underline{v}\left(\underline{x}\right)$ is irrotational and hence can be defined in terms of a potential $\varphi (\underline{x})$. In this regime, we also rigorously demonstrate that the polymer stress must be constant in a cross-section of the thinning thread (far enough from the end-droplets). These results open up the way to obtain the three-dimensional flow field and the free surface shape by using an analogy with electrostatic problems. The result is compared with the universal surface profile obtained in ref. [12] by solving the self-similarity static equations for the neo-Hookean elastic problem (which was shown to nearly coincide with the surface profile obtained numerically based on the Oldroyd-B dynamical equations [12]).

The scope of the paper is as follows: The next Section 2 is summarized in the paragraph just above. The results are generalized to take into account the finite extensibility (FE) of polymer chains (using the FENE-P model) in Section 3. In particular, we establish that the condition, Equation (Equation 1), has to be amended to account for FE, derive the new more general condition, and apply it to rigorously obtain the thinning law for the FENE-P model. The main results are discussed in Section 4 and are briefly summarized in the last section.

## 2. Capillary Thinning for the Oldroyd-B Model

Let us consider the typical setup: a thinning filament between two immobile semi-spherical droplets (Figure 1). The liquid is an unentangled polymer solution. The filament length $L_{f}$ is much larger than its radius a=a(z,t), which slowly decreases over time. This process is driven by capillary forces. It is slow due to the capillary-induced extension of polymer chains (upon the coil–stretch transition) leading to a dramatic increase in the solution viscosity [14]. Five zones can be distinguished in Figure 1: two end-droplets adjacent to solid plates, a nearly cylindrical filament of length $L_{f}$ in the middle, and two transitions zones of length $L_{t}$ (shown in cyan). The whole liquid bridge is assumed to be axially symmetric (around the *z*-axis).

The filament is typically thin, long, and nearly uniform during the polymer extension process [6]:(2)$$L_{f}\gg a_{0}\left(t\right)$$
where $2a_{0}\left(t\right)$ is the thickness in the middle of the thread $\left(z=z_{m}\right)$; $a\left(z,t\right)≃a_{0}\left(t\right)$ in the filament region. As the liquid is incompressible, the uniaxial extension rate in the thread is defined by the thinning rate (note that ${\dot{\epsilon }}_{r}=\partial \left(\ln a_{0}\right)/\partial t$ in Equation (Equation 3) is the (negative) extension rate in the radial direction, and that (by virtue of the incompressibility) $\dot{\epsilon }+2{\dot{\epsilon }}_{r}=0$):(3)$$\dot{\epsilon }=\partial v_{z}/\partial z≃-2\partial \left(\ln a_{0}\right)/\partial t$$
so the axial flow velocity is: $v_{z}≃\dot{\epsilon }\left(z-z_{m}\right)$ in the filament region of length $L_{f}$. Outside this region, the liquid cross-section increases rapidly, leading (by virtue of the incompressibility) to a decrease of $v_{z}$. Thus, the velocity maximum, $v_{0}=\max\left|v_{z}\right|≃\dot{\epsilon }L_{f}/2$, is reached near the filament ends.

Let us turn to the force balance in the transition zones next to the filament ends where the surface shape deviates from a perfect cylinder and its radius shows a significant *z*-dependence. The transition region between the thread and the second droplet is depicted in Figure 2. (Note that the origin, z=0, is located near the righthand end of the thread. We define the point z=0 by the condition $a\left(z=0\right)=k_{e}a_{0}$, where the factor $k_{e}$ can be chosen as convenient; here it was set $k_{e}\approx 1.25$.) As argued below, the axial size, $L_{t}$, of this region (where the droplet surface significantly deviates from both cylindrical and spherical) is relatively small, $L_{t}∼a_{0}\ll L_{f}$.

To describe the flow and the free surface shape, we can benefit from two approximations. First, we neglect the inertial effects assuming that the ‘inertial pressure’ ∼$\rho v^{2}$ is much lower than the capillary pressure γ/a (here, ρ is density and γ is the surface tension of the liquid):(4)$$\rho v_{0}^{2}\ll \gamma /a_{0}$$This is a natural condition since a uniform filament is not stable if inertial effects are significant [3]. Second, we assume that polymer stress $(\sigma_{p}$) dominates over the viscous stress of the solvent: $\sigma_{p}\gg \eta_{s}\left|\frac{\partial v}{\partial z}\right|$. Again, this condition is obvious well inside the thread (as, otherwise, the polymer effect would be rheologically negligible, and the liquid would be equivalent to the Newtonian solvent there); however, it must be *assumed* for the transition region. This assumption is justified if the capillary number $C_{n}$ (defined in Equation (Equation 71)) is small, as discussed in item 5 of Section 4 (see text around Equation (Equation 71)). It is also assumed that the liquid is incompressible:(5)$$\nabla \cdot \underline{v}\equiv v_{\beta ,\beta }=0$$
where $\underline{v}=\underline{v}\left(\underline{x}\right)$ is the velocity field.

The master dynamical equation is then reduced simply to the force balance:(6)$$\sigma_{\alpha \beta ,\beta }=0$$
where α,β are Cartesian components, $\sigma_{\alpha \beta }=\sigma_{\alpha \beta }\left(\underline{x},t\right)$ is the total stress tensor at the point $\underline{x}=\left(x_{1},x_{2},x_{3}\right)$, $x_{3}\equiv z$, and $Y_{,\beta }\equiv \partial Y/\partial x_{\beta }$ is a derivative of a variable *Y* along the coordinate $x_{\beta }$. It involves two contributions due to the pressure field ($p=p(\underline{x},t$)) and the extra polymer stress:(7)$$\sigma_{\alpha \beta }=-p\delta_{\alpha \beta }+\sigma_{\alpha \beta }^{p}$$The polymer stress tensor is [15]:(8)$$\sigma_{\alpha \beta }^{p}=\frac{c}{N}\langle \frac{\partial F_{el}}{\partial R_{\alpha }}R_{\beta }\rangle$$
where $F_{el}=F_{el}\left(\underline{R}\right)$ is the elastic free energy of a polymer chain with end-to-end vector $\underline{R}$. Angular brackets here mean the statistical average, *c* is the concentration of polymer segments (repeat units), and *N* is their number in one chain (a monodisperse polymer is considered in the present paper). For polymer liquids in marginal or theta-solvent conditions [16,17] (and for concentrated polymer solutions) a Gaussian chain model is applicable during the coil–stretch transition when the chains are far from their full extension, R≪L (here, *L* is the chain contour length), so the elastic energy is [15]:(9)$$F_{el}≃\frac{3}{2}T\frac{R^{2}}{R_{coil}^{2}}$$
where *T* is temperature in energy units, $R_{coil}=N^{1/2}b_{s}=\sqrt{Ll_{K}}$ is the unperturbed coil size $(b_{s}$ is the polymer statistical segment, $l_{K}$ is its Kuhn segment). Hence, recalling Equation (Equation 8), we obtain:(10)$$\sigma_{\alpha \beta }^{p}≃3\frac{cT}{N^{2}b_{s}^{2}}\langle R_{\alpha }R_{\beta }\rangle ≃3\frac{cT}{N^{2}b_{s}^{2}}{\bar{R}}_{\alpha }{\bar{R}}_{\beta }$$
where we took into account that all the chains in a fluid element are strongly stretched along the same axis (defined by some unit vector $\underline{m}$), so that the end-to-end vector $\underline{R}$ of most chains is nearly parallel to $\underline{m}$, and therefore, $\langle R_{\alpha }R_{\beta }\rangle ≃const\;m_{\alpha }m_{\beta }=\;{\bar{R}}_{\alpha }{\bar{R}}_{\beta }$, where $\underline{\bar{R}}=\bar{R}\underline{m}$, and $\bar{R}$ is the root-mean-square (rms) of the end-to-end distance. For the sake of simplicity, in what follows, we omit the ‘bar’ over $\underline{R}$ and *R*. (Thus, from now on, *R* is simply the rms end-to-end distance of polymer chains, which is widely used in polymer physics.) Note that we consider the regime of significant chain extension, $R\gg R_{coil}$, since, otherwise, $\sigma_{\alpha \beta }^{p}$ would be comparable with the ideal gas pressure of polymer molecules, which is low for long chains. Note also that $\underline{\bar{R}}=\underline{R}\left(\underline{x},t\right)$ in Equation (Equation 10) is the mean end-to-end vector which generally depends on the chain position $\underline{x}$ and time *t*. (More precisely, $\underline{x}$ is the position of the chain center-of-mass. In all the cases, we assume that the fields such as $\underline{v}\left(\underline{x},t\right)$ or $\underline{R}\left(\underline{x},t\right)$ do not change much on the polymer length-scale $R_{coil}$. This condition is satisfied if $R_{coil}\ll a_{0}$.)

Obviously, well inside the thread, polymer chains must be stretched along its main axis (z): $R_{z}≃R_{0}\gg R_{coil}$, $R_{x_{1}}∼R_{x_{2}}∼R_{coil}$, where $R_{0}$ is $\bar{R}$ (rms of polymer end-to-end distance) in the thread; thus, the only relevant component of the polymer elastic stress tensor is:(11)$$\sigma_{zz}^{p}≃3\frac{cT}{N^{2}b_{s}^{2}}R_{0}^{2}$$(other components being subdominant).

The model described above is virtually equivalent to the Oldroyd-B dumbbell model, as the polymer conformation is defined solely by the end-to-end vector $\underline{R}$, and the polymer stress is quadratic in $\underline{R}$. Using this model, it was established that the thinning of a polymer thread in the viscoelastic regime $(R_{coil}\ll R\ll L$) is quasi-exponential [1,8]:(12)$$a_{0}\propto \exp\left(-\left(1/3\right)t/\tau \right),\;\;R_{0}\propto \exp\left(\left(1/6\right)t/\tau \right)$$
where τ is the stress relaxation time $(\tau =\tau_{p}/2$, where $\tau_{p}$ is the longest relaxation time of a polymer coil), and *t* is the time passed since the beginning of coil stretching.

Equation (Equation 12) (which is applicable in the transient regime during the polymer stretching stage) implies that the thinning rate $\dot{\epsilon }$ (cf. Equation (Equation 3)) is constant, $\dot{\epsilon }≃2/\left(3\tau \right)$. However, strictly speaking, the flow in the thread is not steady, as its radius $a_{0}=a_{0}\left(t\right)$ is decreasing. Nevertheless, the flow in the entrance region (the ‘corner flow’) can be considered at a different angle: The characteristic thinning time is $1/\dot{\epsilon }∼\tau$. Comparing it with the characteristic entering time $\tau_{ent}∼L_{t}/v_{0}∼L_{t}/\left(\dot{\epsilon }L_{f}\right)$, we see that $\dot{\epsilon }\tau_{ent}∼L_{t}/L_{f}∼a_{0}/L_{f}\ll 1$, so:(13)$$\tau_{ent}/\tau \ll 1.$$This means that the entrance flow (at the interface between the thread and a large droplet) is *quasi-stationary* to a good accuracy, since the ratio $a_{0}/L_{f}$ is typically small in experiments, $a_{0}/L_{f}≲0.01$ [3,6,7,18]. Let us emphasize again that it is assumed here, following the previous theoretical studies [1,7,8,11,12], that the thread is perfectly cylindrical, the flow velocity in the thread is parallel to the main axis (z), and the chains are stretched along it. The force balance (cf. Equations (Equation 6) and (Equation 7)) in such a geometry demands that the pressure *p* is constant (independent of both radial and axial coordinates), $p=\gamma /a_{0}$, and that the polymeric stress $\sigma_{p}\equiv \sigma_{zz}^{p}-\sigma_{rr}^{p}≃\sigma_{zz}^{p}$ is *z*-independent. In Section 3, we show that well inside the thread, the stress $\sigma_{p}$ is also constant in the radial direction (cf. text below Equation (Equation 56)). Note that, in contrast, $\sigma_{p}$ does depend on both *z* and *r* in the transition regions.

Since the entrance flow (see Figure 2) can be considered as steady, its time dependence can be neglected at the relevant time scale $\tau_{ent}$. The formal boundary conditions on the free liquid surface, defined by r=a(z), are: the surface must tend to a tube of radius $a_{0}$ at $z/a_{0}\to -\infty$ (here and below, $z/a_{0}\to -\infty$ means that z<0 and $L_{f}\gg \left|z\right|\gg a_{0}$), and to a sphere of large radius (which can be approximated by a flat plane) at $r/a_{0}\to +\infty$. (Note that we use cylindrical coordinates $z=x_{3}$, $r=\sqrt{x_{1}^{2}+x_{2}^{2}}$, as shown in Figure 2.) In the limit of no inertia, the equation of motion is reduced to the local force balance, which can be written as (see Equations (Equation 6), (Equation 7), and (Equation 10)). (Most equations below are approximate, just like Equation (Equation 10). However, the ‘=’ sign is used there for simplicity, except where it is important to emphasize their approximate nature.)
(14)$$3\frac{cT}{N^{2}b_{s}^{2}}{\left(R_{\alpha }R_{\beta }\right)}_{{}_{,\beta }}=p_{,\alpha }$$

The pressure at the free surface is defined by its curvature *C*:(15)$$p=\gamma C=\frac{\gamma }{\sqrt{1+{\left(a_{,z}\right)}^{2}}}\left[\frac{1}{a}-\frac{a_{,zz}}{1+{\left(a_{,z}\right)}^{2}}\right]\;\mathrm{at}r=a\left(z\right)$$
where
(16)$$a\left(z\right)\to a_{0}\;\mathrm{at}z/a_{0}\to -\infty ,\;a_{,zz}\equiv \frac{\partial^{2}a}{\partial z^{2}}\to 0\;\mathrm{at}z/a_{0}\to +\infty$$
and *p* is the excess pressure on the top of atmospheric pressure. The flow is therefore defined by an interplay of capillary and viscoelastic effects.

The basic Equations (Equation 5) and (Equation 14) involve two unknown fields: polymer extension, $\underline{R}=\underline{R}\left(\underline{x}\right)$, and velocity $\underline{v}=\underline{v}\left(\underline{x}\right)$. However, in the entrance zone, these fields can be easily related to each other. The idea is that the polymer chains must be extended along the stationary streamlines, and their fragments (blobs) must simply follow the stream (i.e., a blob at point $\underline{x}$ moves with the velocity $\underline{v}\left(\underline{x}\right)$). In fact, the relative motions of the blobs due to elastic forces can be neglected as being too slow since the relevant polymer relaxation time $(\tau_{p}$) is much longer than $\tau_{ent}$ (see Equation (Equation 13)). As a result, the chain extension vector $\underline{R}$ stays nearly proportional to the flow velocity $\underline{v}$ for kinematic reasons:(17)$$\underline{R}\left(\underline{x}\right)≃B\underline{v}\left(\underline{x}\right)$$

For a steady flow, the above equation follows from the kinematic equations $d\underline{v}/dt\equiv \partial \underline{v}/\partial t+\underline{v}\cdot \nabla \underline{v}=\underline{v}\cdot \nabla \underline{v}$, $d\underline{R}/dt=\underline{R}\cdot \nabla \underline{v}$ (cf. Equation (Equation 80) with $1/\tau_{p}\to 0$), which ensure that if $\underline{R}=B\underline{v}$ at some point (which is the case for z<0, $a_{0}\ll \left|z\right|\ll L_{f}$), the latter equation must stay valid (with B=const) along the whole streamline in the transition region where the rate $1/\tau_{p}$ can be neglected. Here, d/dt means the full rate of change ‘following the fluid’ for a material element.

Of note, Equation (Equation 17) remains valid in the *extended* transition region, including the adjacent end part of the thread at z<0, $\left|z\right|<\Delta ,$ where $a_{0}\ll \Delta \ll L_{f}$. The condition $\Delta \ll L_{f}$ ensures that the transition time (along the whole transition region including the Δ-part) is much shorter than the polymer relaxation time $\tau_{p}$, so that the polymer relaxation process can be neglected in the Δ-part. On the other hand, the condition $a_{0}\ll \Delta$ is sufficient to state that at −z≳Δ the thread is nearly perfectly cylindrical, so at z=−Δ its radius is $a≃a_{0}$, the rms chain extension is $R≃R_{0}$ and the flow velocity is $v≃v_{0}$ also being uniform in the cross-section. Then, applying Equation (Equation 17) at z=−Δ we obtain
$$B≃R_{0}/v_{0}$$
and therefore *B* is the same for all the streamlines. Equations (Equation 10) and (Equation 17) obviously agree with the convected Jeffreys (or Oldroyd-B) models with infinite relaxation time [14] (cf. item 11 of the Discussion).

The force balance Equation (Equation 14) then becomes:(18)$$p_{,\alpha }=\tilde{A}v_{\alpha ,\beta }v_{\beta }$$
where
(19)$$\tilde{A}=3\frac{cT}{N^{2}b_{s}^{2}}\frac{R_{0}^{2}}{v_{0}^{2}}$$
and we took into account the flow incompressibility from Equation (Equation 5). Equations (Equation 18) and (Equation 5), together with the boundary conditions in Equation (Equation 15), an obvious condition $v_{n}=0$ at the free surface (where $v_{n}$ is the flow velocity component normal to the surface), and
(20)$$v\to v_{0}\;\mathrm{at}z/a_{0}\to -\infty ,\;p\to 0\;\mathrm{at}z/a_{0}\to +\infty$$
completely define the flow, the pressure field, and the shape of the free surface in the entrance zone. (Recall that $z/a_{0}\to -\infty$ physically means that z<0, $a_{0}\ll \left|z\right|\ll L_{f}$; in a similar fashion, $z/a_{0}\to +\infty$ means $z\gg a_{0}$. Note also that we consider the limit of a very large droplet, so the excess pressure deeply inside it must tend to 0.) Obviously, Equations (Equation 15) and (Equation 16) also imply that:(21)$$p\to p_{0}=\gamma /a_{0}\;\mathrm{at}z/a_{0}\to -\infty$$

There is only one essential non-dimensional parameter in the problem: the ratio $X\equiv \sigma_{p}a_{0}/\gamma$ of polymer stress $\sigma_{p}$ to the capillary pressure well inside the filament, $p_{0}=\gamma /a_{0}$. It was set to X=1 in ref. [8], while X=2 was derived in ref. [12]. (Note that the quantity $X_{f}=\frac{1+X}{2}$ was denoted *X* in ref. [5]; $X_{f}$ was defined there as the ratio of the net tensile force in the thread to $2\pi \gamma a_{0}$. As follows from Table I of ref. [5], the parameter $X=2X_{f}-1$ is as important as $X_{f}$ itself for Newtonian liquids considered there. This parameter (X) is even more important for polymer liquids as it highlights a relation between the main quantities of interest, the polymer stress $\sigma_{p}$, and the capillary pressure γ/a.) Below, we show that *X* can be easily obtained based on the equations considered above. To this end, let us first analyze the pressure *p* variation along a streamline. Equation (Equation 18) gives (with $v\equiv \left|\underline{v}\right|$ and the coordinate *u* equal to the curvilinear distance along the streamline):(22)$$\partial p/\partial u=\left(\tilde{A}/v\right)v_{\alpha ,\beta }v_{\alpha }v_{\beta }=\tilde{A}v\partial v/\partial u$$
leading to $\partial p/\partial u=\left(\tilde{A}/2\right)\partial v^{2}/\partial u$; hence,
(23)$$p-\frac{\tilde{A}}{2}v^{2}≃const\;\;\mathrm{along}\mathrm{streamline}$$This equation is applicable in the entrance zone, where the velocity is changing fast along streamlines. It connects the local pressure and velocity just like the Bernoulli’s principle does for an ideal fluid. We shall refer to Equation (Equation 23) as the anti-Bernoulli law (due to the ‘minus’ sign leading to a higher pressure for a faster flow). The const in Equation (Equation 23) can be easily found for the flow of Figure 2: Recall that each streamline is coming from the uniform thread, where both *p* and $\underline{v}$ are constant over a cross-section. Therefore,
(24)$$p-\frac{\tilde{A}}{2}v^{2}=const=0$$
in the whole entrance zone $(\left|z\right|\ll L_{f}$). Note that Equations (Equation 24) and (Equation 23) are valid if the Newtonian solvent friction is negligible as assumed below Equation (Equation 4) and as discussed in item 5 of the Discussion section (cf. Equation (Equation 71)). The rhs in the above equation equals 0 since both *p* and $\underline{v}$ must vanish deeply in the droplet region at $z\gg a_{0}$ (cf. Equation (Equation 20)). Thus, Equations (Equation 20), (Equation 21), and (Equation 24) imply that
(25)$$p_{0}=\frac{\tilde{A}}{2}v_{0}^{2},\;\;v_{0}^{2}=\frac{2\gamma }{\tilde{A}}\frac{1}{a_{0}}$$
and we obtain $\sigma_{p}$ in the thread using Equations (Equation 11) and (Equation 19):(26)$$\sigma_{p}=\tilde{A}v_{0}^{2}=2\gamma /a_{0}.$$Hence, the parameter *X* introduced below Equation (Equation 21) is:(27)$$X\equiv \sigma_{p}/p_{0}=\sigma_{p}a_{0}/\gamma =2.$$The total thread tension force T (defining the total momentum flux through its cross-section) is therefore:(28)$$T=2\pi a_{0}\gamma +\pi a_{0}^{2}\left(\sigma_{p}-p_{0}\right)=3\pi a_{0}\gamma$$The two terms in the middle of the above equation represent the surface and the internal axial stress contributions to the total tension T. Equations (Equation 27) and (Equation 28) agree with the results of ref. [12].

It is worthwhile to note that Equations (Equation 17)–(Equation 24) are valid for a virtually arbitrary flow geometry (involving a long channel as a part of it). Using Equations (Equation 18) and (Equation 24), we obtain:(29)$$\left(v_{\beta ,\alpha }-v_{\alpha ,\beta }\right)v_{\beta }\equiv 0$$Let us now take into account that, typically, the experimental setup is such that both the free surface and the flow are axisymmetric. This symmetry is already implied in Equation (Equation 15). For the flow velocity, it means that its axial and radial components $(v_{z}$ and $v_{r}$) do not depend on the polar angle θ, while the polar velocity component $v_{\theta }$ must vanish (here, we use cylindrical coordinates (z,r,θ), cf. Figure 2):(30)$$v_{z}=v_{z}\left(r,z\right),\;\;v_{r}=v_{r}\left(r,z\right),\;\;v_{\theta }\equiv 0$$The factor in brackets in Equation (Equation 29) is related to the vorticity $\underline{\omega }=\nabla \times \underline{v}$
(31)$$v_{\beta ,\alpha }-v_{\alpha ,\beta }=\epsilon_{\alpha \beta \gamma }\omega_{\gamma }$$
where $\epsilon_{\alpha \beta \gamma }$ is the Levi-Civita symbol. Therefore, Equation (Equation 29) is equivalent to:(32)$$\underline{\omega }\times \underline{v}\equiv 0$$For an axisymmetric flow, $\underline{\omega }$ is always parallel to the azimuthal direction $(\omega_{r}=\omega_{z}=0$); hence, $\underline{\omega }$ and $\underline{v}$ are orthogonal. Equation (Equation 32) then gives ωv≡0, so:(33)$$\nabla \times \underline{v}\equiv 0$$
everywhere except at the stagnation points (where $\underline{v}=0$), which are not expected in the entrance zone. Therefore, the flow is irrotational:(34)$$\underline{v}=\nabla \varphi$$
where φ=φ(r,z) is a potential field which must satisfy the Laplace equation (in view of Equation (Equation 5)):(35)$$\nabla^{2}\varphi =0$$Furthermore, on using Equations (Equation 24) and (Equation 25), the boundary condition, Equation (Equation 15), transforms to:(36)$$v^{2}=\frac{v_{0}^{2}a_{0}}{\sqrt{1+{\left(a_{,z}\right)}^{2}}}\left[\frac{1}{a}-\frac{a_{,zz}}{1+{\left(a_{,z}\right)}^{2}}\right]\;\mathrm{at}r=a\left(z\right)$$
while another condition at the free surface $\left(v_{n}=0\right)$ simply says that the boundary curve r=a(z) must be a streamline with $r\to a_{0}$ at $z/a_{0}\to -\infty$ (obviously, the axisymmetric problem is essentially two-dimensional, so we can set θ=0 and $r=x_{1}$ without loss of generality).

Further statements, which can facilitate numerical solution of the above equations for the velocity field $\underline{v}\left(\underline{x}\right)$, can be proven in an elementary way. At large $z\gg a_{0}$, the flow is asymptotically similar to the electrostatic field of an electric charge: (37)$$\underline{v}≃\frac{Q\underline{x}}{r^{3}},\;\;Q=\frac{1}{2}v_{0}a_{0}^{2}$$
where $\underline{x}$ is the position vector, $r=\left|\underline{x}\right|$ is the distance from the origin, and *Q* is obtained based on the incoming volume flow rate, which is equal to $\pi a_{0}^{2}v_{0}$. Equation (Equation 37) implies that at $r\gg a_{0}$, the pressure decreases as:(38)$$p/p_{0}≃\frac{1}{4}{\left(\frac{a_{0}}{r}\right)}^{4}$$The pressure therefore becomes negligible in the droplet far from the entrance point, so the droplet shape there must be close to the static shape defined solely by the Laplace pressure. It leads to the condition of constant mean curvature, C→0, in the case of large droplet, so [4]:(39)r(z)≃acosh(z/a)forz≫a
where $a∼a_{0}$. The above equation is consistent with Equation (3).18 of ref. [12].

Next, let us note that the structures of Equations (Equation 35) and (Equation 36) are such that a simple rescaling $r\to r/a_{0}$, $z\to z/a_{0}$, and $v\to v/v_{0}$ makes the problem free of any parameters (in other words, the solution is self-similar). It is therefore enough to solve the above equations for $a_{0}=1$, $v_{0}=1$ and Q=1/2. The unique rescaled flow field and the free surface curve a(z) can be found numerically in two steps: (1) by setting a trial a(z) and then solving the Laplace Equation (Equation 35) in the region $z>-z_{max}$, $r<r_{max}$, and r<a(z) (see Figure 2) with Neumann boundary conditions: $\nabla \varphi \cdot \underline{n}=0$ at the free surface (r=a(z)), $\nabla \varphi \cdot \underline{n}=1$ in the filament cross-section at $z=-z_{max}$, and $\nabla \varphi \cdot \underline{n}=const\;$ at the hemisphere $r=r_{max}$ inside the droplet (here $\underline{n}$ is unit vector normal to the surface); (2) by adjusting a(z) iteratively so as to satisfy the anti-Bernoulli equation ${\left(\nabla \varphi \right)}^{2}=\frac{1}{\sqrt{1+{\left(a^{\prime }\right)}^{2}}}\left[\frac{1}{a}-\frac{a^{\prime \prime }}{1+{\left(a^{\prime }\right)}^{2}}\right]$ at r=a(z).

The resultant thread shape and streamlines in the entrance zone (obtained for $z_{max}=6$ and $r_{max}=30$ taken to obtain a well-converged solution to an accuracy of ∼0.5%) are shown in Figure 3. In the thread region, z→−∞, the free surface tends to a cylindrical shape in an exponential fashion:(40)$$a\left(z\right)≃1+const\;e^{kz},\;-z\gg 1$$
where k≈1.393. (The constant k>0 is the lowest root of the characteristic equation $2kJ_{0}\left(k\right)=\left(1+k^{2}\right)J_{1}\left(k\right)$, which comes from the main correction to the potential field φ at −z≫1: $\varphi -\varphi_{0}≃const\;J_{0}\left(kr\right)e^{-kz}$, where $\varphi_{0}=z$ corresponds to a perfectly uniform flow. Here $J_{0},$$J_{1}$ are Bessel functions.)

The asymptotic behavior in the opposite regime, z≫1, can be deduced from the force balance, which says that the total momentum flux in the cylindrical part (the thread tension T=3πγ, cf. Equation (Equation 28)) must be equal to the momentum flux, $T_{+}$, across a hemisphere r=const≫1, z>0. The contribution of both polymer stress and pressure to $T_{+}$ can be neglected since they rapidly tend to zero at large r: $\sigma_{p}∼p\propto 1/r^{4}$ (cf. Equation (Equation 38)); hence, $pr^{2}\to 0$ at r→∞. Therefore, $T_{+}$ at r≫1 is defined solely by the surface tension γ:(41)$$T_{+}≃2\pi a\gamma /\sqrt{1+{\left(a^{\prime }\right)}^{2}}$$The equation $T_{+}=3\pi \gamma$ leads to
(42)$$a\left(z\right)\propto \exp\left(2z/3\right),\;\;z\gg 1$$This asymptotic behavior was predicted [12] using a similar argument. However, the point that the polymer stress can be neglected in the droplet region was not proven there. Of note, the numerical solution for a(z) shown in Figure 3 is in harmony with the asymptotic laws, Equations (Equation 40) and (Equation 42).

## 3. Finite Extensibility Effects

### 3.1. FENE-P Model

The theory described above was developed for the Oldroyd-B model. Below, it is generalized to account for nonlinear finite extensibility effects. In the general case, the polymer chain tension $\frac{\partial F_{el}}{\partial R}$ depends on the end-to-end distance *R* in a nonlinear fashion (cf. Equation (Equation 9)):(43)$$\frac{\partial F_{el}}{\partial R}=\frac{T}{R_{1}^{2}}R\kappa_{FE}\left(s\right)$$
where s=R/L is the stretching degree (whose maximum value is 1), and $R_{1}^{2}=Nb_{s}^{2}/3$ is related to the coil size $R_{coil}=b_{s}\sqrt{N}$. The factor $\kappa_{FE}\left(s\right)$ must tend to 1 at low *s*, when the Gaussian model is valid (recall that we consider polymers in marginal or theta solvents). By contrast, $\kappa_{FE}$ must generally strongly increase for 1−s≪1 to provide a finite extensibility (FE): $\kappa_{FE}\to \infty$ at s→1. The whole function $\kappa_{FE}\left(s\right)$ is, however, not universal: it depends on the polymer flexibility mechanism. One of the most popular FE models was introduced by Warner [19]. It serves as a part of the FENE-P dumbbell model [14,20,21] with
(44)$$\kappa_{FE}\left(s\right)=\frac{1}{1-s^{2}}$$Using Equations (Equation 43) and (Equation 44), we obtain:(45)$$F_{el}=-\frac{3}{2}TN_{K}\ln\left(1-s^{2}\right)$$
where $N_{K}=L/l_{K}=L^{2}/\left(Nb_{s}^{2}\right)$ is the number of Kuhn segments in a polymer chain.

The elastic force, Equation (Equation 43), defines the polymer stress tensor, cf. Equation (Equation 8):(46)$$\sigma_{\alpha \beta }^{p}=\frac{c}{N}\frac{\partial F_{el}}{\partial R_{\alpha }}R_{\beta }=\frac{G}{R_{1}^{2}}\kappa_{FE}\left(s\right)R_{\alpha }R_{\beta }$$
where G=cT/N is the shear elastic modulus defining the polymer elastic response at short times (t≲τ). (Note that internal relaxation modes are ignored in the FENE models.)

### 3.2. Equations of Motion in the Entrance Zone

We now turn to dynamical equations applicable in the transition zone where the flow is non-uniform but is virtually steady (cf. Equation (Equation 13) and the text below it). In this regime, as argued above Equation (Equation 17), the polymer extension (the end-to-end vector) $\underline{R}$ is proportional to the flow velocity $\underline{v}$, cf. Equation (Equation 17). Note that polymer contraction due to elastic force can be neglected in the entrance region even if the chains are stretched close to full extension (s close to 1) because *s* rapidly decreases hydrodynamically (affinely with the fluid element) down to s∼0.5 upon entering the transition region due to a fast deceleration of the flow there. Indeed, the rate of affine contraction of chains in the transition region is defined by the flow velocity gradient, $\left|\partial v_{z}/\partial z\right|∼v_{0}/a_{0}∼\left(L_{f}/a_{0}\right)\tau^{-1}$; it is much faster than the rate (∼1/τ) for the non-Gaussian elastic relaxation, since the factor $L_{f}/a_{0}$ (the ratio of the thread length to its radius) is very large, as stated in Equation (Equation 2) (and as observed experimentally). Therefore, considering the entrance flow for the FENE-P model, we can treat the flow as quasi-stationary setting τ→∞ and using Equation (Equation 17) just like for the Oldroyd-B model.

The factor *B* in Equation (Equation 17) is constant along a streamline, but strictly speaking, it may vary among the streamlines. In Section 2, we provided a plausible argument showing that *B* is constant in the whole entrance zone. This property is proved below on more general grounds. Remarkably, even with a variable *B* equation, $\underline{R}=B\underline{v}$ still ensures that the field $\underline{R}\left(\underline{x}\right)$ is solenoidal:(47)$$\nabla \cdot \underline{R}=B\nabla \cdot \underline{v}+\underline{v}\cdot \nabla B=0$$
since $\underline{v}$ and ∇B are necessarily orthogonal.

Using Equations (Equation 6), (Equation 7), (Equation 46), and (Equation 47) we obtain:(48)$$\frac{\partial p}{\partial x_{\alpha }}=\frac{c}{N}\frac{\partial^{2}F_{el}}{\partial R_{\alpha }\partial R_{\gamma }}R_{\beta }\frac{\partial R_{\gamma }}{\partial x_{\beta }}$$Recall that we neglect inertial effects and the stress contribution due to the solvent viscosity for the reasons considered in Section 2. Next, taking into account that $F_{el}$ can be considered as a function of $R^{2}$, one obtains:(49)$$\frac{\partial^{2}F_{el}}{\partial R_{\alpha }\partial R_{\gamma }}=2F_{el}^{\prime }\delta_{\alpha \gamma }+4F_{el}^{\prime \prime }R_{\alpha }R_{\gamma }$$
where
(50)$$F_{el}^{\prime }\equiv \frac{\partial F_{el}}{\partial \left(R^{2}\right)},\;F_{el}^{\prime \prime }\equiv \frac{\partial^{2}F_{el}}{\partial {\left(R^{2}\right)}^{2}}$$The above equations lead to:(51)$$\frac{\partial p}{\partial x_{\alpha }}=2\frac{c}{N}\left[F_{el}^{\prime }R_{\beta }\frac{\partial R_{\alpha }}{\partial x_{\beta }}+F_{el}^{\prime \prime }R_{\alpha }R_{\beta }\frac{\partial R^{2}}{\partial x_{\beta }}\right]$$Considering the pressure gradient along a streamline, using Equation (Equation 51), we obtain:(52)$$\frac{\partial p}{\partial u}=\frac{\partial F}{\partial u}$$
where the function F=F(R) must satisfy the equation:(53)$$\frac{\partial F}{\partial u}=\frac{c}{N}\left(F_{el}^{\prime }+2F_{el}^{\prime \prime }R^{2}\right)\frac{\partial R^{2}}{\partial u}$$
and *u* is a curvilinear coordinate (the contour length) along a streamline. From Equation (Equation 53), we obtain:(54)$$F\left(R\right)=\frac{c}{N}\left(2F_{el}^{\prime }R^{2}-F_{el}\right)$$
so that Equation (Equation 52) leads to:(55)p−F(R)=const
along each streamline. Taking into account that F(0)=0 and that both *p* and the chain extension *R* must vanish in the droplet far away from the thread, we find that const=0 in Equation (Equation 55), so:(56)p=F(R)Equation (Equation 56) is valid in the whole entrance region including the thread-end region (z<0,$a\ll \left|z\right|\ll L_{f}$), where $p≃p_{0}$ and $v_{z}≃v_{0}$ are nearly uniform both axially and radially (in a cross-section). Using Equation (Equation 56), we find that the same must be true for the chain extension *R* and the polymer stress $\sigma_{p}$ related to it. Recalling that both $\sigma_{p}$ and $R=R_{0}$ are *z*-independent within the thread (see text below Equation (Equation 13)), we conclude that these quantities do not change in the radial direction in the whole thread. This means, in particular, that the factor $B=R_{0}/v_{0}$ in equation $\underline{R}=B\underline{v}$ (cf. Equation (Equation 17)) is the same for all streamlines. Equation (Equation 56) can be therefore written in the form:(57)p=F(Bv)
which generalizes the anti-Bernoulli law, Equation (Equation 24).

For the FENE model, Equations (Equation 45) and (Equation 54) give:(58)$$F\left(R\right)=\frac{3}{2}GN_{K}\tilde{F}\left(s\right)$$
where R=sL and
(59)$$\tilde{F}\left(s\right)=\frac{2s^{2}}{1-s^{2}}+\ln\left(1-s^{2}\right)$$In the regime s≪1, both F(R) and the elastic energy density $F_{el}\left(R\right)=\frac{c}{N}F_{el}\left(R\right)$ are nearly quadratic in *R* and are equal:(60)$$F\left(R\right)≃F_{el}\left(R\right)≃\frac{1}{2}G\frac{R^{2}}{R_{1}^{2}}=\frac{3}{2}GN_{K}s^{2}\;,\;s\ll 1$$Therefore, in this case, the anti-Bernoulli law also says that the local pressure is equal to the elastic energy density. However, in the opposite regime of nearly fully stretched chains, 1−s≪1, F(R) becomes much larger than $F_{el}\left(R\right)$:$$F\left(R\right)≃\frac{3}{2}GN_{K}\frac{1}{1-s},\;F_{el}\left(R\right)≃\frac{3}{2}GN_{K}\ln\frac{1}{1-s}\;,\;1-s\ll 1$$

Based on the anti-Bernoulli law, we have shown in Section 2 that the flow must be irrotational for the Oldroyd-B model: $\underline{\omega }\equiv \nabla \times \underline{v}=0$ (cf. Equation (Equation 33)). This property is not preserved in the more general case considered in this section: it is replaced by a more general relation derived below (cf. Equation (Equation 63)).

For symmetry reasons, $\underline{\omega }$ has only one nonzero component (=ω) in the azimuthal direction perpendicular to the rz-plane. Using local coordinates with axis *u* tangential to a streamline (at a given point *O*) and axis $u_{⊥}$ perpendicular to it (cf. Figure 4), we obtain:(61)$$\omega =\partial v/\partial u_{⊥}-\partial v_{⊥}/\partial u$$
where $v_{⊥}$ is the velocity component along axis $u_{⊥}$. Noting that $\partial v_{⊥}/\partial u$ is proportional to the local curvature, *C*, of the streamline:(62)$$\partial v_{⊥}/\partial u=Cv$$
and using Equations (Equation 54) and (Equation 56), we find:$$\frac{\partial p}{\partial x_{\alpha }}=\frac{\partial F}{\partial R^{2}}\frac{\partial R^{2}}{\partial x_{\alpha }}=\frac{c}{N}\left(F_{el}^{\prime }+2F_{el}^{\prime \prime }R^{2}\right)\frac{\partial R^{2}}{\partial x_{\alpha }}$$Comparing the above equation with Equation (Equation 51) and recalling Equation (Equation 17), we finally obtain:(63)$$\left(F_{el}^{\prime }+2F_{el}^{\prime \prime }R^{2}\right)\frac{\partial v}{\partial u_{⊥}}=F_{el}^{\prime }\frac{\partial v_{⊥}}{\partial u}$$Hence, in the general case, $\partial v/\partial u_{⊥}\neq \partial v_{⊥}/\partial u$ and ω≠0. Note, however, that Equations (Equation 62) and (Equation 63) still ensure that $\underline{\omega }=0$ for a straight streamline, C=0. This is true, in particular, both well inside the droplet and in the thread. Therefore, Equations (Equation 37) and (Equation 38) remain valid at $r\gg a_{0}$.

### 3.3. Flow in the Thread and the Thinning Law

Turning to the flow in the cylindrical part of the thread, we first recall that it is uniform in a cross-section, so both the pressure *p* and the polymer stress $\sigma_{\alpha \beta }^{p}$ do not depend either on the axial coordinate *z* or *r* in the thread. The pressure $p=p_{0}$ in the thread is defined by Equation (Equation 56), where $R=R_{0}$ is the chain extension:(64)$$p_{0}=F\left(R_{0}\right)$$On the other hand, the polymer stress $\sigma_{p}=\sigma_{zz}^{p}-\sigma_{rr}^{p}$ is (cf. Equation (Equation 46)):(65)$$\sigma_{p}=3GN_{K}s^{2}\kappa_{FE}\left(s\right),\;\;s\equiv R_{0}/L$$The *s*-dependence of the ratio $X=\sigma_{p}/p_{0}$ is shown in Figure 5. It is clear that $\sigma_{p}/p_{0}$ is not a constant: for s≪1, the above equations lead to $\sigma_{p}/p_{0}=2$, while for 1−s≪1, the result is $\sigma_{p}/p_{0}=1$.

The kinetics of the chain extension for the FENE-P model can be described with evolution equation:(66)$$\frac{dR}{dt}=\dot{\epsilon }R-\frac{1}{\tau_{p}}\kappa_{FE}\left(R/L\right)R$$Here, $\dot{\epsilon }$ is the extension rate in the axial direction, which is related to the thread thinning rate, see Equation (Equation 3), and $R=R_{0}$. The second term in the rhs of Equation (Equation 66) is proportional to the elastic force [20]. (Note that a thermal noise term is not present in Equation (Equation 66) since thermal fluctuations are negligible in the regime $R\gg R_{coil}$ we consider.)

Next, using Equations (Equation 3), (Equation 21), and (Equation 64), we find that $\dot{\epsilon }=2\frac{d\ln\tilde{F}\left(s\right)}{dt}$, so Equation (Equation 66) becomes:(67)$$\frac{1}{R}\frac{dR}{dt}=2\frac{d\ln\tilde{F}\left(s\right)}{dt}-\frac{1}{\tau_{p}}\kappa_{FE}\left(s\right)$$It leads (using Equation (Equation 44)) to the following ODE for s=s(t):(68)$$\frac{ds^{2}}{dt}=4s^{2}\frac{d\ln\tilde{F}\left(s\right)}{dt}-\frac{1}{\tau }\frac{s^{2}}{1-s^{2}}$$Equation (Equation 68) is valid for the FENE-P model. It can be integrated to give (with substitution $y\equiv s^{2}$):(69)$$-\ln y+y+4\int \frac{1+y}{2y+(1-y)\ln(1-y)}dy=\frac{t}{\tau }+const$$The above equation defines the time dependence of the stretching degree, s(t), which is shown in Figure 6. The time dependence of the ratio $\sigma_{p}/p_{0}$ is indicated there as well. Obviously, this ratio strongly decreases near the breakup point (the const in Equation (Equation 69) was chosen to obtain the breakup at t=0). The resultant thinning law, $a_{0}\left(t\right)=\gamma /F\left(R_{0}\right)=\frac{2\gamma }{3GN_{K}}\frac{1}{\tilde{F}\left(s\right)}$, is drawn in Figure 7. The straight dashed line highlights the fact that $a_{0}\left(t\right)$ follows the classical exponential law, $a_{0}\left(t\right)\propto \exp\left(-\frac{t}{3\tau }\right)$,[1]], as long as $s=R_{0}/L≲0.5$.

## 4. Discussion

1. In the present paper, we investigated the dynamics of a viscoelastic liquid bridge (a cylindrical ligament) connecting two large droplets (see Figure 1). Its thinning is governed by capillary and viscoelastic forces and is coupled to the flow both in the ligament and in the transition zone between the thread and a droplet (see Figure 2). As a fluid, we consider an unentangled polymer solution whose rheology can be described by either the Oldroyd-B or the FENE-P dumbbell models. The developed theory is focused on the regime of strongly stretched polymers whose conformational tensor $\langle R_{\alpha }R_{\beta }\rangle$ is dominated by a single eigenvalue (here, $\underline{R}$ is the end-to-end vector of a polymer chain). The models we used assume that the stress tensor contribution of a polymer chain is defined by the vector $\underline{R}$ and therefore is proportional to $R_{\alpha }R_{\beta }$. The prefactor in such a relation is either constant (cf. Equation (Equation 10) for Oldroyd-B model) or depends on the degree of chain stretching s=R/L (for the FENE-P model). Both models ignore hydrodynamic interactions (HDI) in the system (see point 9 below).

As one of the main results, we have shown that for the Oldroyd-B model, the capillary-driven flow is nearly irrotational and satisfies the Laplace equation. In addition, we have established a general relation between the pressure and velocity fields (valid for a broad class of polymer flows) which is referred to as the anti-Bernoulli law (cf. Equation (Equation 23); an equivalent static conservation law was derived for a neo-Hookean elastic solid [12]). Using these results, we obtained a self-similar solution (involving a single length-scale, the radius $a_{0}$ of the thread) for the shape of the free surface and for the pressure and velocity fields. In cylindrical coordinates (r,z), the free surface is given by $r=a\left(z\right)=a_{0}f\left(z/a_{0}\right)$. The numerically obtained free surface profile a(z) for $a_{0}=1$ and the corresponding flow structure are shown in Figure 3. The calculated a(z) is compared in Figure 8 with the similarity solution for the bead-string structure of a neo-Hookean elastic body obtained in ref. [12]. (Note that $l_{e}=2^{1/3}a_{0}$ was taken as a unit length in ref. [12], so the data have been rescaled accordingly.) The obvious good agreement supports the conclusion of refs. [12,13] that (as conjectured earlier in refs. [22,23]) the neo-Hookean model asymptotically reproduces the geometry of a polymer liquid bridge formed in the course of its capillary thinning (according to the Oldroyd-B constitutive equation) in the regime of sufficiently long polymer relaxation time τ. Of note, it was recently shown that the universal interfacial shapes obtained in ref. [12] (and, therefore, the theoretical profile of Figure 3 and Figure 8) are in agreement with high-resolution experimental data on aqueous solutions of high-molecular-weight polymers (PEO) and biopolymers (hyaluronic acid) [24,25].

2. The conservation (anti-Bernoulli) law defining pressure in terms of the flow field (cf. Equation (Equation 24)) was obtained here based on the steady-flow dynamical equations (Equations (Equation 6) and (Equation 17)). Of note, in the form of Equation (Equation 56), p=F(R), this law is akin to the conservation law established for neo-Hookean solids [12]. The main difference is that Equation (Equation 56) is more general: it also accounts for non-linear elastic effects related to the finite extensibility of polymer chains.

3. To quantitatively establish the thinning kinetics, $a_{0}=a_{0}\left(t\right)$, one needs a ‘closure’ relation between the pressure $p_{0}$ and the axial polymer stress $\sigma_{p}$ in the thread. This relation cannot be obtained by considering a cylindrical thread as such. Long ago, Entov and Hinch [8] assumed that the total axial stress in the thread $(\sigma_{p}-p_{0}$) must be equal to that in the droplet, i.e., to zero (if atmospheric pressure is subtracted from $p_{0}$): $p_{0}=\sigma_{p}$. It is found here that the rigorous anti-Bernoulli law, Equation (Equation 24), demands a different relation, $p_{0}=\sigma_{p}/2$, in agreement with more recent predictions for the Oldroyd-B model [7,11,12]. It is remarkable that a generalization of the anti-Bernoulli law for polymer chains with finite extensibility, Equation (Equation 57), leads to a variable ratio $X=\sigma_{p}/p_{0}$ which depends on the degree of polymer extension s=R/L. While for s≪1 we obtain $\sigma_{p}/p_{0}=2$, this ratio decreases with *s* (see Figure 5), reaching its minimum, $\sigma_{p}/p_{0}=1$, in the regime of completely stretched chains, s→1. Incidentally, it is the latter ratio that was assumed in ref. [8].

It may seem that the model of a nearly uniform thread cannot be used for *s* close to 1 since in this regime of highly extended chains the liquid behaves nearly as a Newtonian fluid with renormalized viscosity [4,26,27,28] $\eta^{*}≃\eta_{s}\left(1+\frac{\pi }{18k_{H}}\frac{c}{N}L^{3}\right)$, where $k_{H}≃0.5\ln(1/\phi$) is a hydrodynamic factor [29] and ϕ is volume fraction of polymer. Thus, the classical Plateau–Rayleigh instability (undulations of the thread shape) must emerge in this regime (at *s* close to 1, $s>s_{c}∼0.5$), so that the uniform thread approximation seems to fail. The point, however, is that: (i) Before the instability (at $s<s_{c}$), the undulations are weak, and their amplitude (in radial direction) due to thermal fluctuations is ${\tilde{a}}_{0}∼\sqrt{T/\gamma }$. (Here we make a plausible assumption that the amplitude of thermal fluctuations of the radius in the dynamically stable thread are comparable to that for a statically stable liquid cylinder of the same diameter.) Therefore, ${\tilde{a}}_{0}\ll a_{0}$, since, typically, $\sqrt{T/\gamma }≲1\;\mathrm{nm}$, while $a_{0}\gg 1\;\mathrm{nm}$. (ii) The growth rate of their amplitude (coming from the Rayleigh theory [30]) is $\left|d(\ln\tilde{a})/dt\right|∼\frac{1}{6}\frac{\gamma }{a\eta^{*}}$, so it is comparable with the thread *thinning* rate, [4,5] $\dot{\epsilon }/2∼\frac{1}{6}\frac{\gamma }{a\eta^{*}}$. Therefore, as the thread thins by a factor of *k*, the undulations grow by the same factor, so the thread radius *a* becomes comparable with the undulation amplitude $\tilde{a}$ for $k=k^{*}∼\sqrt{a_{0}/{\tilde{a}}_{0}}\gg 1$. Considering a=a(s) as a function of *s*, we find, according to the theory of Section 3.3 that a(s)/a(0.5)∼1−s for s≳0.5, that is, 1/k∼1−s. The above argument therefore shows that the thread undulations remain small ($\tilde{a}\ll a$) if $1-s\gg 1/k^{*}$. Hence, the theory of Section 3.3 is actually applicable in the regime where the chain extension degree *s* is close to the maximum (=1): this is true in the region $1\gg 1-s\gg 1/k^{*}$ since $k^{*}\gg 1$. In this regime, the ratio *X* indeed becomes very close to 1, as stated in the previous paragraph.

4. The developed theory was applied to obtain the thread thinning dynamics for the FENE-P model. While this problem was treated already in ref. [8], it is treated here for the first time in a rigorous way. The numerically obtained thinning law, $a_{0}\left(t\right)$ (see Figure 7) includes both the elasto-capillary (exponential thinning) and the terminal stage (quasi-linear thinning). It is based on the asymptotically exact dependence of the ratio $X=\sigma_{p}/p_{0}$ on the degree of chain stretching established in Section 3 (see also point 3 above).

The predicted thinning law is universal (up to arbitrary horizontal and vertical shifts) when plotted in semilog scale vs. t/τ. The theory is compared with experimental data on semidilute, unentangled polymer solutions (cf. Figure 5 of ref. [18]) in Figure 9, where t/τ=0 for the theoretical curves is set to be the putative breakup point $\left(a_{0}=0\right)$. One can observe a good agreement down to the crossover regime; however, at later times (on the right to a red circle), the experimental curve deviates from the prediction. We attribute this discrepancy to a new regime most probably associated with the blistering (pearling) instability [2,31], which is normally accompanied by the polymer/solvent phase separation and formation of secondary solvent droplets (which can be both flow- and capillary-induced [28,29,32,33,34]). Interestingly, according to Figure 9, this transition occurs where the chains are not yet fully stretched, at s≈0.5, corresponding to $X=\sigma_{p}/p_{0}\approx 1.75$.

5. Considering the thinning of a cylindrical polymer thread, we assumed that the pressure *p* there is constant in the radial direction (i.e., it is homogeneous in a cross-section). This assumption was adopted in the previous theoretical studies on capillary thinning [1,7,8,11,12]. In the case of negligible inertia, it simply follows from the force balance in the cylindrical geometry since the flow velocity $\underline{v}$ is parallel to the main axis, and so is the elastic force due to stretched polymer chains. In addition, however, the theory developed in Section 2 and Section 3 implies that the axial polymer stress $\sigma_{p}$ is also uniform (independent of *r*) in the thread. For the Oldroyd-B model, this condition is justified in the limit of vanishing solvent viscosity as follows: First, it is straightforward to show that the argument (cf. Section 2) leading to the anti-Bernoulli law, Equation (Equation 24), is generally valid in the form:(70)$$p=\sigma_{∥}^{p}/2$$(where $\sigma_{∥}^{p}=\sigma_{uu}^{p}$ is the polymer stress component along the streamline). Obviously, in the thread, $\sigma_{∥}^{p}≃\sigma_{p}$ and (as argued above) p=const. Therefore, Equation (Equation 70) implies that $\sigma_{p}$ is also constant (r,z-independent) there. The conjecture that for negligible solvent viscosity the stress profile in the thread is uniform [12] is thus rigorously proved using a dynamical approach. (Of note, it was also established [12] that the radial distribution of the axial stress is uniform for a soft neo-Hookean elastic solid. Hence, here, again, we confirm the correspondence between viscoelastic and elastic models [13].) In Section 3.2, we show that this statement is also valid for the models with finite chain extensibility.

Numerical studies of the Oldroyd-B model with a finite Newtonian solvent viscosity $\eta_{s}$ [12] reveal a weak radial dependence of $\sigma_{p}$, which was attributed to an effect of $\eta_{s}$. We tend to agree with this interpretation. The effect of $\eta_{s}$ (which was neglected in the present study) is unimportant as long as $\eta_{s}\left|\frac{\partial v}{\partial z}\right|\ll \sigma_{p}$, which is true everywhere (including the entrance region) if a capillary number is small:(71)$$C_{n}\equiv \eta_{s}L_{f}/\left(6\tau \gamma \right)\ll 1$$The above condition was adopted in the previous sections. Note that for any $\eta_{s}$, this number $(C_{n}$) can be made however small, provided that the polymer time τ is sufficiently long. For a finite $\eta_{s}$, the polymer stress $\sigma_{p}$ in the thread may vary along the radius. We expect that (at least for a small $C_{n}$) a maximal relative variation of $\sigma_{p}$ in the radial direction should be defined by $C_{n}$. It may also be worth mentioning that an increase in the solvent’s viscosity does not necessarily lead to a higher $C_{n}$ since the ratio $\eta_{s}/\tau$ depends mainly on the polymer molecular weight rather than on $\eta_{s}$ (note that τ is nearly proportional to $\eta_{s}$ if polymer volume fraction is low [15,35]).

6. It is well-known [7,11,12,36] that the elasto-capillary thread thinning (according to an exponential law, Equation (Equation 12)) is preceded by a fast initial process of the thread formation, which is primarily governed by capillary and inertial forces. In a typical scenario, a cylinder of initial radius $R_{0}$ and length $>2\pi R_{0}$ undergoes the Plateau instability. A thread of radius $a_{0i}≃R_{0}{\left(\frac{GR_{0}}{2\gamma }\right)}^{1/3}$ is formed as a result of this process [7,11,12,36]. The characteristic time of this initial stage is inertial in nature, $t_{i}∼{\left(\rho R_{0}^{3}/\gamma \right)}^{1/2}$. In the inertial regime, $t_{i}$ also defines the period of oscillations of the emerging semi-droplets (connected by the thread). Their damping rate is proportional to the solvent viscosity $\eta_{s}$ defining the dissipation rate, so the damping time is $t_{damp}∼t_{i}^{2}\gamma /\left(\eta_{s}R_{0}\right)∼\rho R_{0}^{2}/\eta_{s}$. (Note that $t_{damp}\gg t_{i}$ as long as $R_{0}^{2}\gg \eta_{s}^{2}/\left(\rho \gamma \right)$, which literally means that initially the system falls in the inertial regime.) This time must not exceed τ (since, otherwise, the thread thinning would be strongly perturbed by droplet oscillations). Hence, we have to demand that
(72)$$\rho R_{0}^{2}/\left(\eta_{s}\tau \right)≲1$$This condition is compatible with Equation (Equation 71) if
(73)$$\rho R_{0}^{3}/\left(\tau^{2}\gamma \right)\ll 1$$
which simply means that $t_{i}\ll \tau$. Note that the condition (Equation 73) also ensures that inertial effects are negligible in the thread: $\rho v_{0}^{2}\ll \gamma /a_{0}$, as was assumed in Section 2 (cf. Equation (Equation 4)). Thus, Equations (Equation 71) and (Equation 72) define the range of allowed $\eta_{s}$:(74)$$\rho R_{0}^{2}/\tau ≲\eta_{s}\ll 2\tau \gamma /R_{0}$$(Here, we assumed that the filament length is comparable with $R_{0}$, $L_{f}∼3R_{0}$, which was the case in the previous numerical studies [7,12,23].) Obviously the conditions of Equation (Equation 74) are always satisfied for a sufficiently long polymer relaxation time τ.

7. The fluid dynamics considered here are dominated by the longitudinal polymer stress $\sigma_{∥}^{p}$ (along a streamline). This property is quite natural in the regime of strongly stretched polymer chains, when the polymer chain end-to-end distance *R* is much larger than the equilibrium coil size $R_{coil}$, $R\gg R_{coil}$. In fact, it was implicitly assumed that polymer chains are strongly stretched during an initial fast inertio-capillary stage of polymer thread development [36], i.e., before the elasto-capillary regime of the thread thinning considered in this paper. Such initial process leads to a uniaxial polymer stress field, cf. Equation (Equation 10), both in the thread (of radius $a_{0}$) and in the droplet/thread transition region of size $∼a_{0}$. However, the applicability of this approximation in the droplet at large distances from the transition region, $r\gg a_{0}$, may be questioned. Indeed, $R\propto v\propto 1/r^{2}$ (cf. Equations (Equation 17) and (Equation 37)) strongly decreases there with the distance r, so the chains become less stretched in the radial direction (along the streamline), and, more importantly, they become expanded in the lateral directions: the lateral size $R_{⊥}$ grows as $R_{⊥}∼R_{coil}r/a_{0}$. It leads to an emergence of the lateral stress $\sigma_{⊥}∼G{\left(r/a_{0}\right)}^{2}$. The stress growth stops as soon as the travel time $t_{v}\left(r\right)$ for a fluid element to reach the distance r from the origin becomes comparable with the stress relaxation time τ. Using Equation (Equation 37) with $v_{0}=\dot{\epsilon }L_{f}/2=L_{f}/\left(3\tau \right)$, we obtain $t_{v}\left(r\right)∼r^{3}/\left(v_{0}a_{0}^{2}\right)$, so that $t_{v}∼\tau$ is reached at $r=r^{*}∼L_{f}^{1/3}a_{0}^{2/3}$. The maximum lateral stress is, therefore, $\sigma_{⊥}∼G{\left(L_{f}/a_{0}\right)}^{2/3}$. This stress should lead to an additional curvature $\Delta C∼\sigma_{⊥}/\gamma$ of the free surface, which can be neglected if ΔCr≪1, leading to the condition:(75)$$E_{c}\equiv GL_{f}/\left(3\gamma \right)\ll 1$$Here, $E_{c}$ is nearly equivalent to the elasto-capillary number introduced in refs. [7,12] (with $L_{f}∼3R_{0}$). It is noteworthy that the same condition, $E_{c}\ll 1$, also ensures that the chains are strongly stretched during the initial inertio-capillary process: $R\gg R_{coil}$, $a_{0}\ll L_{f}$ (cf. refs. [7,12]).

8. To sum up the previous two points, there are three main conditions of validity of the theory: Equations (Equation 71), (Equation 74), and (Equation 75). It is noteworthy that Equations (Equation 71) and (Equation 75) are similar and are actually equivalent in the dilute/semidilute transition regime since the polymer contribution $\eta_{p}$ to the total viscosity is $\eta_{p}=G\tau$ (for the Oldroyd-B model) and $\eta_{p}∼\eta_{s}$ at $c∼c^{*}$, where $c^{*}$ is the coil overlap concentration. Furthermore, in the semidilute regime $(c>c^{*}$), Equation (Equation 71) follows from Equation (Equation 75).

9. In the present study, we totally neglected the hydrodynamic interactions (HDI), which are not incorporated either in the Oldroyd-B or FENE-P models. Such a simplification, which was also adopted in the previous theoretical works on the subject [1,7,8,11,12,37], may be appropriate in the semidilute or concentrated solution regime $(c>c^{*}$). The effects related to topological constraints (entanglements between polymer chains) [38] are disregarded here as well, so the region of applicability should be formally restricted to the unentangled polymer regime. It is worth noting, however, that both models employed here seem to also work well for dilute polymer solutions, at least as far as the capillary thinning kinetics are concerned [8,28,32,39]. Moreover, the main effect of the HDI is to modify the polymer relaxation time, which is completely irrelevant in the entrance zone with high deformation rate. Therefore, the approach to describe the flow in such zones (cf. Section 2 and Section 3.2) also remains valid for dilute polymer solutions.

10. In this paper, we ignored any effects of polymer/solvent separation assuming that polymer concentration c=const. It is well-known that an inhomogeneous polymer distribution in a solution may be generated by a flow, for example, due to the stress–concentration coupling (SCC) effect [32,40,41,42]. This effect is not relevant for the cylindrical thread where the stress is uniform, so the concentration is uniform as well. Moreover, while in principle, the SCC effect is present in the entrance zone, it is very weak there: in fact, we argued (cf. Section 2) that polymer chains are extended along the streamlines which are curved in the entrance zone. This curvature generally leads to a net elastic force perpendicular to the streamline, $f_{⊥}$, acting on an internal part of the chain, $f_{⊥}∼f_{el}R/a$, where *R* is the chain length, 1/a is the typical streamline curvature, and $f_{el}=\partial F_{el}/\partial R$ is the elastic tension of the chain (cf. Equation (Equation 9)). The force, $f_{⊥}$, leads to the perpendicular velocity, $v_{⊥}$, of the chain relative to the solvent, $v_{⊥}∼\left(R/\tau \right)\left(R/a\right)$. The typical time the chain spends in the entrance zone is t∼a/v, so its typical lateral displacement is $u_{⊥}∼v_{⊥}t∼\left(R/\tau \right)\left(R/v\right)$. The relative concentration change due to this displacement is $∼u_{⊥}/a∼R^{2}/\left(av\tau \right)∼R^{2}/\left(aL_{f}\right)$, where we recalled that $v∼L_{f}\dot{\epsilon }∼L_{f}/\tau$. Taking also into account that $R\ll a\ll L_{f}$, we find that $u_{⊥}/a\ll 1$, so the SCC effect is indeed negligible and c≃const in the system we considered.

11. The polymer stress tensor defined in Equation (Equation 8), with the conditions specified before Equation (Equation 10), can be written as:(76)$$\sigma_{\alpha \beta }^{p}≃3\frac{cT}{N^{2}b_{s}^{2}}\langle R_{\alpha }R_{\beta }\rangle$$Its rate-of-change prescribed by the Oldroyd-B model can be expressed in terms of the upper convective derivative:(77)$$\overset{\nabla p}{\sigma_{\alpha \beta }}=d\sigma_{\alpha \beta }^{p}/dt-v_{\alpha ,\gamma }\sigma_{\gamma \beta }^{p}-\sigma_{\alpha \gamma }^{p}v_{\beta ,\gamma }$$
as
(78)$$\tau \overset{\nabla p}{\sigma_{\alpha \beta }}=-\sigma_{\alpha \beta }^{p}+\delta_{\alpha \beta }cT/N$$(here, $d/dt\equiv \partial /\partial t+\underline{v}\cdot \nabla$ is the full rate-of-change referred to a moving physical element of the fluid). The above equation is equivalent to the standard Oldroyd-B equation (see, e.g., Equation (2.2) in ref. [12]). (Note that $\sigma_{\alpha \beta }^{p}$ here is the full polymer stress, while the polymer stress $\sigma_{\left(p\right)}$ in the standard Oldroyd-B model does not include the isotropic equilibrium polymer stress contribution $\delta_{\alpha \beta }cT/N$, so $\sigma_{\alpha \beta }^{p}=\sigma_{(p)\alpha \beta }+\delta_{\alpha \beta }cT/N$.)

To further simplify the equations, we took into account that, in the thread (and, in particular, near its mid-point), all the chains are strongly stretched in a similar way along the filament axis (z-axis). Therefore, the polymer stress tensor there must be dominated by a single eigenvalue corresponding to an eigenvector $\underline{R}$ parallel to this axis, as argued around Equation (Equation 10):(79)$$\sigma_{\alpha \beta }^{p}≃3\frac{cT}{N^{2}b_{s}^{2}}R_{\alpha }R_{\beta }$$The prefactor in the above equation is such that it makes sure that $\left|\underline{R}\right|$ is equal to the root-mean-square (rms) end-to-end distance of polymer chains in a fluid element. Equations (Equation 77) and (Equation 78) then ensure that if Equation (Equation 79) is valid at some point (note that Equations (Equation 76) and (Equation 79) together are equivalent to Equation (Equation 10)), it will be also valid at all the points downstream (as long as the polymer chains remain strongly stretched in a single direction), with vector $\underline{R}$ changing according to
(80)$$\frac{dR_{\alpha }}{dt}=v_{\alpha ,\beta }R_{\beta }-\frac{1}{\tau_{p}}R_{\alpha }$$
where $\tau_{p}=2\tau$. To derive Equation (Equation 80), we neglected the second term in the rhs of Equation (Equation 78): this term is small because $\left|\sigma^{p}\right|\gg cT/N$ since the polymer (the dumbbell) is strongly stretched.

In the case of a steady flow, it is enough to know that Equation (Equation 10) is applicable in a cross-section of the thread: then, it must also be valid in the whole volume downstream (as long as the eigenvalue of the tensor $\underline{\underline{\sigma }}$${}^{p}$ associated with the eigenvector $\underline{R}$ remains strongly dominating). Note also that (again for a steady flow) the full rate of change of velocity is:(81)$$\frac{dv_{\alpha }}{dt}=v_{\alpha ,\beta }v_{\beta }$$Interestingly, Equations (Equation 80) and (Equation 81) show that if $\underline{R}=B\underline{v}$ at some point at t=0, the two vectors remain parallel at all points downstream (along the streamline which includes the initial point): $\underline{R}=B\left(t\right)\underline{v}$ with $B\left(t\right)=B\left(0\right)\exp\left(-t/\tau_{p}\right)$. The validity of this statement can be demonstrated simply by substitution of the solution, $\underline{R}=B\left(t\right)\underline{v}$, in Equation (Equation 80) to make sure that this equation is satisfied using Equation (Equation 81).

## 5. Summary

We studied the thinning dynamics of a liquid bridge containing long flexible polymer chains in the viscoelastic (elasto-capillary) regime where the thread diameter decreases according to the classical exponential law and the chains are highly stretched with respect to their equilibrium coil size, so that the polymer stress tensor can be approximated by a vector dyad, $\sigma_{\alpha \beta }^{p}\propto R_{\alpha }R_{\beta }$. It is shown that the liquid flow in the transition zone between the thread and a large end-droplet can be considered as quasi-stationary and that the time spent by a polymer chain in this zone is much shorter than the polymer relaxation time τ. Moreover, it is rigorously demonstrated based on the full Oldroyd-B equations (cf. item 11 of the Discussion) that if the polymer stress tensor in a material element is a vector dyad initially, it will remain a dyad provided that the polymer relaxation time is very long, τ→∞. This statement is valid for any flow, whether it is irrotational or not, steady or not, and for any initial orientation of $\underline{R}$. If, in addition, the flow is steady and initially the polymer end-to-end vector $\underline{R}$ is parallel to the velocity $\underline{v}$, then $\underline{R}$ will stay parallel to $\underline{v}$ and proportional to it at all points down the streamline. The validity of Equation (Equation 17) *in the transition zone* is justified this way.

Using the Oldroyd-B model in the transition zone, we established a general relation between the pressure and the flow velocity for the case of negligible solvent viscosity $\eta_{s}$. The relation says that the excess pressure is proportional to the square of velocity. Using this relation (termed the anti-Bernoulli law) and the axial symmetry of the flow, we also found that the flow must be irrotational. These results allow us to obtain the flow field and the free surface shape in the transition zone using an obvious electrostatic analogy (cf. Figure 3). The obtained surface profile is in good agreement with asymptotically exact Equations (Equation 42) and (Equation 40) and with recent numerical results for a similar model [12] obtained with a different theoretical approach (cf. Figure 8).

Using the developed theory, we show that the ratio of the polymer normal stress difference $\sigma_{p}$ to the capillary pressure $p_{0}$ in the thread is $X=\sigma_{p}/p_{0}=2$ in the elasto-capillary regime if the polymer relaxation time τ is sufficiently long (more precisely, when both capillary numbers, $C_{n}$ and $E_{c}$, are small, cf. Equations (Equation 71) and (Equation 75)), which is in agreement with recent theoretical studies [11,12]. Furthermore, it is also proven that $\sigma_{p}$ is uniform in a cross-section of the thread, $\sigma_{p}=const\;$, in the limit $\eta_{s}\to 0$ or for a long polymer relaxation time, τ→∞.

The proposed theory (including the anti-Bernoulli law, cf. Equation (Equation 57)) is also generalized to account for the case of polymer chains with finite extensibility (the FENE-P dumbbell model). It shows (cf. Section 3) that the thread thinning turns significantly faster than the classical exponential law, i.e., Equation (Equation 12), if the degree of chain stretching s=R/L exceeds s∼0.4. In this regime, the maximum flow velocity $v_{0}$ considerably increases with respect to the classical prediction $v_{0}=L_{f}/\left(3\tau \right)$, while the ratio $X=\sigma_{p}/p_{0}$ decreases down to X≃1.

The developed theory paves the way to consider other flow regimes (such as a flow out of a droplet into a filament) and other rheological effects, for example, the effects of solvent viscosity $\eta_{s}$ or of an increasing filament length $L_{f}$ for the capillary thinning dynamics. We expect that the exponential thinning law, Equation (Equation 12), should be significantly modified if $\eta_{s}/\tau ≳\gamma /L_{f}$ (i.e., $C_{n}≳1$). In this regime, the solvent viscosity $\eta_{s}$ must be very important for the transition regions near the thread ends. Such effect was considered to some extent based on the Onsager variational principle [11,37]. However, the effect of $\eta_{s}$ on the flow field in the transition regions was not analyzed in these studies. This effect could be a subject of a separate publication.

## Figures and Tables

**Figure 1 polymers-14-04420-f001:**
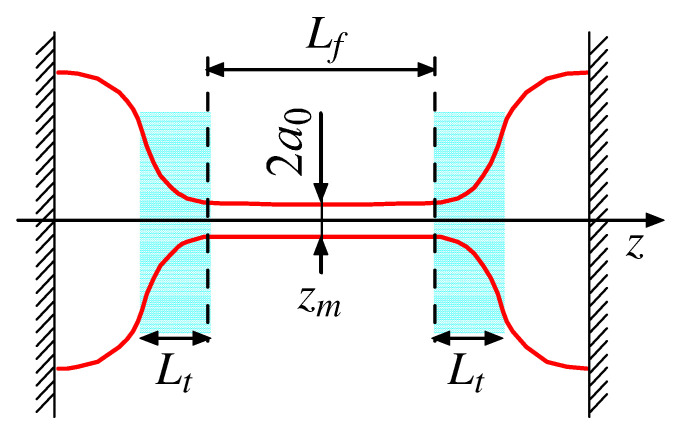
A thinning filament of radius $a_{0}$ and length $L_{f}$ connecting two semi-spherical droplets (a cross-section along the main axis is shown here); $z_{m}$ corresponds to the middle of the thread. $L_{t}\ll L_{f}$ is the size of the filament/droplet transition regions (shown in cyan).

**Figure 2 polymers-14-04420-f002:**
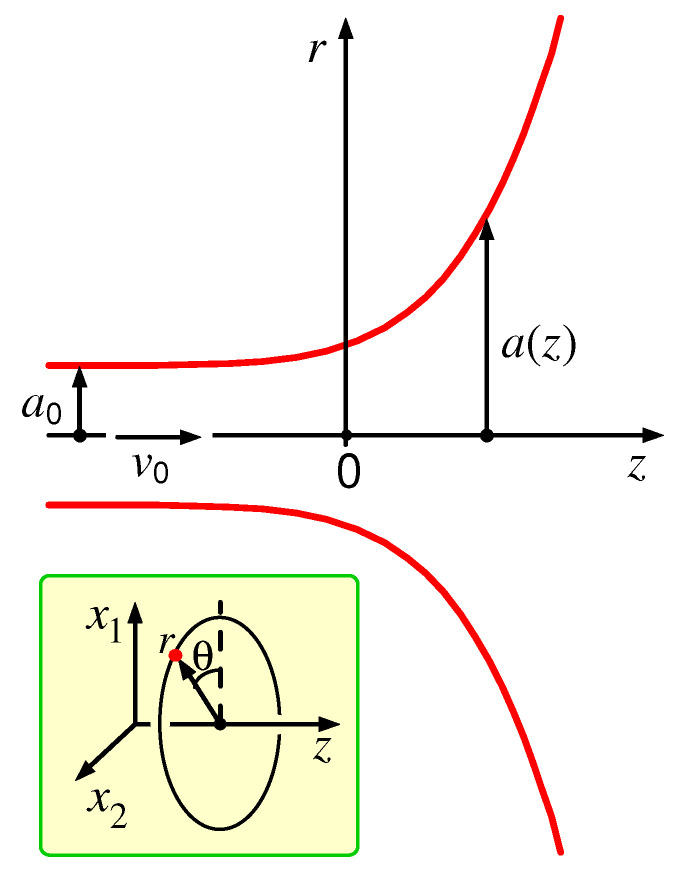
The thread/droplet transition region; r=a(z) defines the free surface in cylindrical coordinates (z,r,θ); $a_{0}$ and $v_{0}$ are the thread radius and the flow velocity at the end of the uniform thread region. Note that a longitudinal cross-section including the cylindrical symmetry axis (z) is shown here. Thus, the upper curve corresponds to the cylindrical angle θ=0, the lower curve to $\theta =180^{\circ }$, and the *r*-axis coincides with the Cartesian $x_{1}$-axis. The inset clarifies the cylindrical coordinates used here.

**Figure 3 polymers-14-04420-f003:**
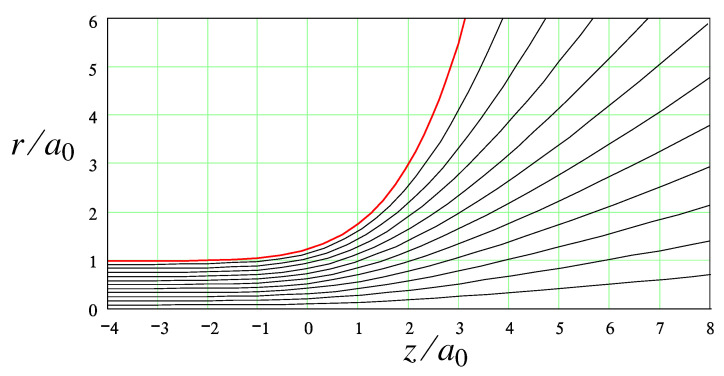
The free surface shape (red line) and streamlines (black curves) in the transition zone coming from Equations (Equation 5), (Equation 33), and (Equation 36) for $a_{0}=1$.

**Figure 4 polymers-14-04420-f004:**
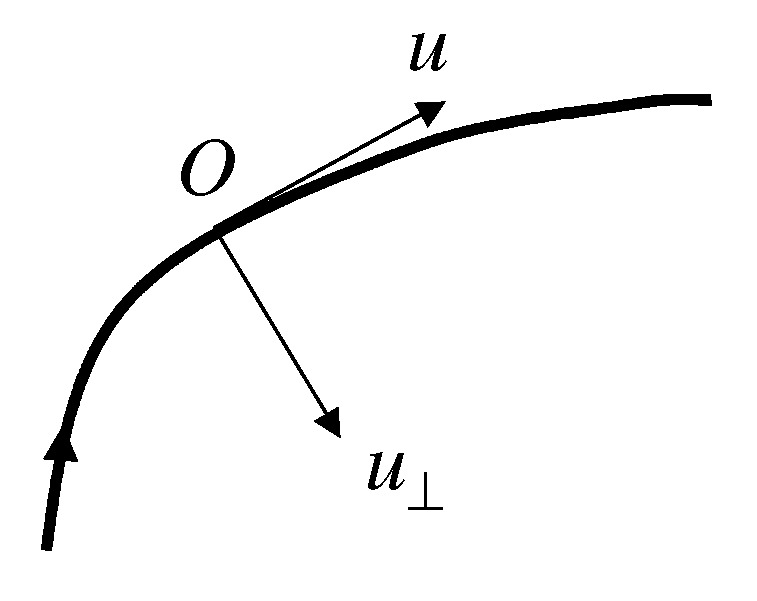
Coordinates *u* and $u_{⊥}$ attached to point *O* of a generic streamline shown as thick curve. Note that some actual streamlines are depicted in Figure 3.

**Figure 5 polymers-14-04420-f005:**
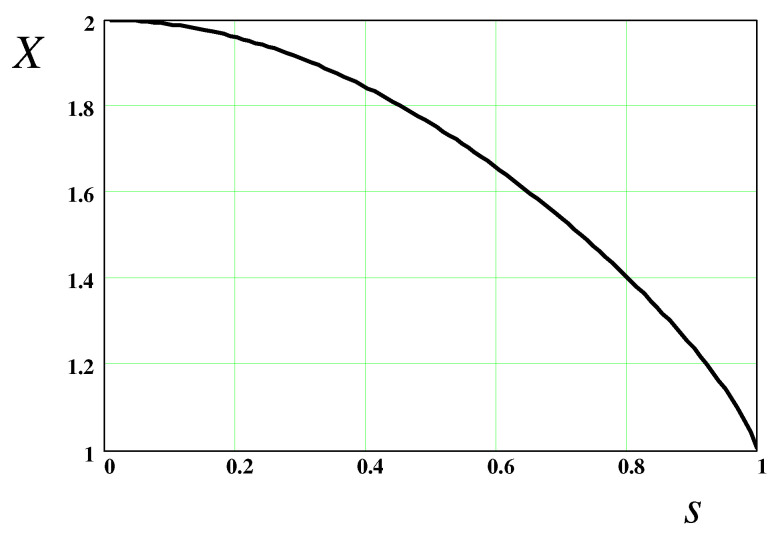
The dependence of $X=\sigma_{p}/p_{0}$ on the stretching degree s=R/L.

**Figure 6 polymers-14-04420-f006:**
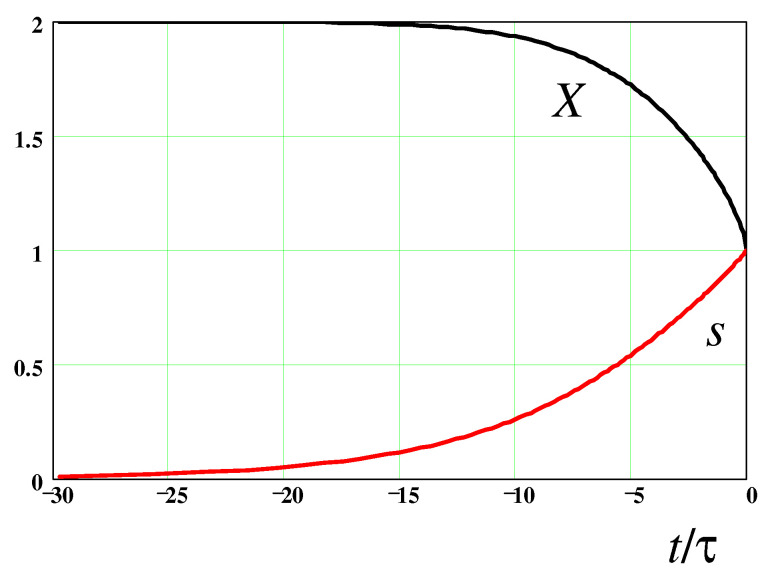
Time dependencies of *X* and *s* (vs. t/τ); t=0 is the putative breakup point.

**Figure 7 polymers-14-04420-f007:**
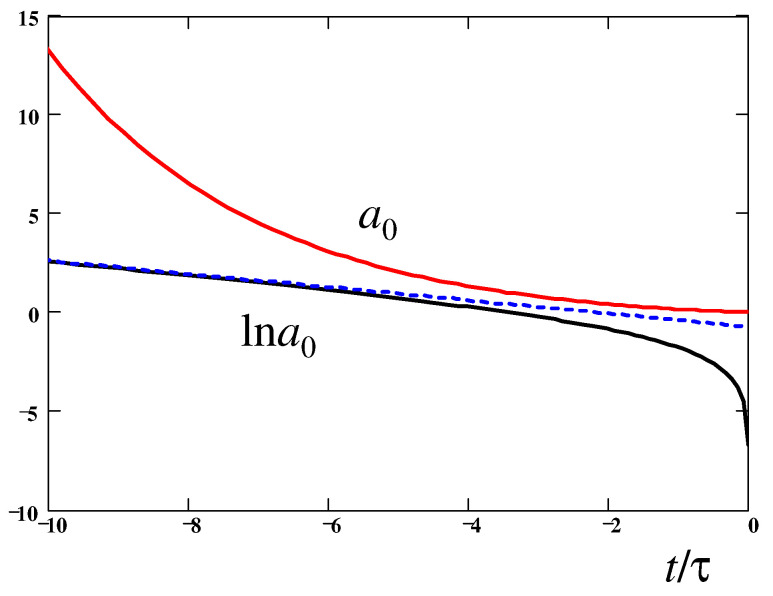
Time-dependence of the thread radius: $a_{0}$ (upper solid curve) and $\ln a_{0}$ (lower curve) vs. t/τ. The dashed line indicates the asymptotic exponential law, Equation (Equation 12).

**Figure 8 polymers-14-04420-f008:**
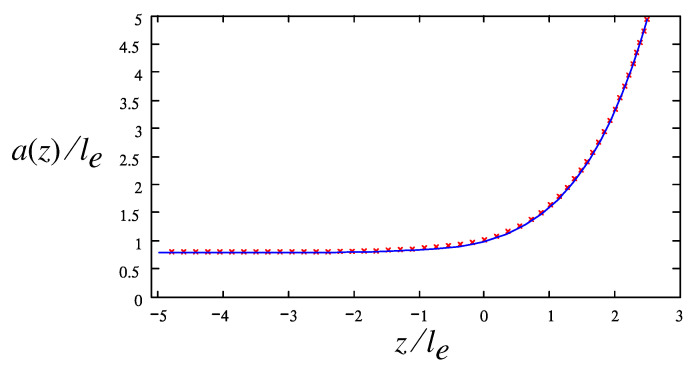
The dependence $a\left(z\right)/l_{e}$ vs. $z/l_{e}$ in the transition zone: our data (crosses), the similarity solution for the bead-string structure obtained using a neo-Hookean elastic model [12] (solid line); $l_{e}=2^{1/3}a_{0}$ is the elasto-capillary length.

**Figure 9 polymers-14-04420-f009:**
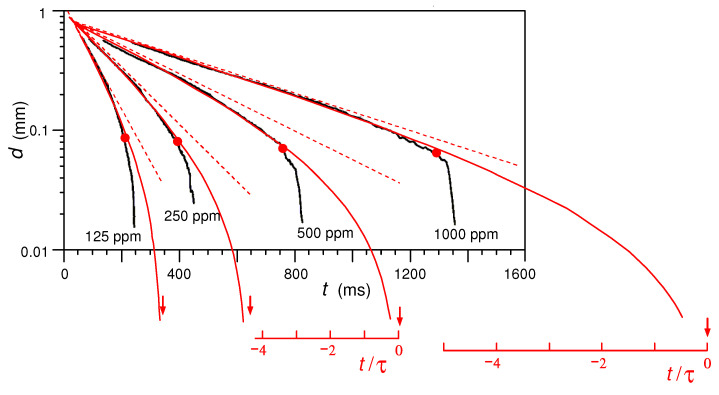
The time dependence of the thread thickness $d=2a_{0}$. Red solid curves: theoretical results (cf. Section 3.3). Thick black curves: experimental data for semidilute aqueous solutions of Praestol-2540 with different concentrations from 125 to 1000 ppm (cf. Figure 5 of ref. [18]). Red circles indicate the onset of significant deviations between theoretical and experimental curves. The theoretical breakup points are indicated with red arrows for each red curve. For the two highest concentrations, we also indicate the reduced time t/τ (with t=0 corresponding to the theoretical breakup). Thin red dashed lines indicate the exponential asymptotic behavior at short times (according to Equation (Equation 12)).

## Data Availability

Not applicable.

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
