# Peer review of "Capillary Thinning of Viscoelastic Threads of Unentangled Polymer Solutions"

_polymers, 2022, doi:10.3390/polym14204420_

Round 1
Reviewer 1 Report
In this paper, authors extend and improve the theory behind the thinning dynamics of a polymer liquid bridging two droplets. Using their new approach, they make a rigorous derivation of the equations governing that phenomenon and are able to reproduce several experimental results. In addition, the paper contains a detailed and useful explanation of the physics and the mathematics employed to develop their approach as well as a revision of previous approaches. The paper is clearly and correctly written. In my opinion, it meets the requirements to be published in Polymers as it is.
Author Response
We thank the reviewer for reading our manuscript and their positive comments.
Reviewer 2 Report
In this work the authors explained, derived, and analyze the flow and filament geometry obtained in the so-called elasto-capillary stretching between two drops for infinite extensible and limited-extensible polymer chains. The analysis is very revealing for people (like me) whose knowledge about filament stretching, in particular when the solvent viscosity plays no role, is poor. The paper is very well written, and I only have few comments and minor corrections; after all, the text goes on explaining by itself all the concerns that were appearing during my reading to the first part of the paper.
1. So that the flow is irrotational is interesting. I always thought that in an extensional deformation, flow has some shearing because motion is more intense on the surface but zero exactly at the extensional axis (radial distance=0) due to incompressibility. But this is not the case when polymer governs the flow and the extension of the thread.
2. When solvent viscosity is negligible but the polymer extensibility is finite (FENE model), then the flow can have some shearing because polymer located at the extensional axis (r=0) reach the maximum extension first than the polymer closer to the surface (that is, the end-to-end distance is a function of the radial distance).
Please, if the above conclusions or my own are incorrect, that means that I picked up incorrectly the message of the paper and I need some clarification from the authors. Perhaps the assumption of very long relaxation times of the polymer cures all these doubts.
Minor comments:
1. I think that reference to figure 8 comes first than figure 7
2. I believe that the thin black line and red dashed line in figure 9 are not explained anywhere in the text
Author Response
We thank the reviewer for careful reading of our ms. and useful comments. Concerning their points:
Point 1. The motion (the velocity magnitude) is indeed stronger near the surface than at the axis in the transition zone. At the axis the radial velocity is of course vanishing, but the axial velocity remains significant. Still the velocity magnitude increases away from the axis in the direction normal to the streamline. However, in the case considered in the ms. (when solvent viscosity is negligible and the flow is governed by polymer stress) this sort of shearing is exactly compensated by the streamline curvature leading to zero net vorticity.
Point 2. In the case of finite chain extensibility the velocity gradient
normal to streamlines decreases significantly and vanishes as the maximum extension is approached (as follows from eq. 63 in the ms.), so the net vorticity emerges due to streamline curvature alone. Noteworthily, while for infinitely extensible chains their extension is higher near the surface than at the axis, with finite extensibility this difference disappears gradually.
Minor points:
1: Figure 8 is now inserted before Figure 7, and their numbers are exchanged.
2: We amend both Fig. 9 and its caption to explain all the lines.
Reviewer 3 Report
Polymers 1931432
Capillary Thinning of Viscoelastic Threads of Unentangled Polymer Solutions
This paper studies the thinning dynamics of a polymer solution between two droplets, which form the so-called beads-on-string structure, in the framework of the Oldroyd-B model developing new analysis techniques such as the anti-Bernoulli law. The results include new findings such as the irrotational velocity field leading to a Laplace equation and those consistent with reported theoretical and experimental data. Many illustrations and curves are helpful for readers to understand the contents. I recommend publication after a minor revision. The comments are as follows:
Comments for the authors:
(1) The differential equation in Eq.(3) includes the ln function. It is helpful to readers if there is a piece of more detailed information on its assumption.
(2) Below Eq.(3), the symbol \Delta is used for z; however, the text and the figures provide no info on \Delta. Readers must stop reading in case no data is provided at the start.
(3) In Eq. (30), a three-dimensional cylindrical coordinate is used by referencing Fig.2. However, Fig. 2 is written on a two-dimensional plane. A three-dimensional illustration of the velocity and vorticity is helpful for readers to understand the result directly or intuitively.
(4) It is helpful for readers if there includes a relation between the solid curve in Fig. 4 and the free surface in Fig.2 or curves in Fig.3.
(5) In the final paragraph of the Summary, the authors wrote some expectations on modifying the exponential thinning formula. If this expectation is the future topic of study, it should be written clearly. Then, the implication of the paragraph becomes more apparent.
Author Response
We thank the reviewer for careful reading of our manuscript and useful points made. Our response is given below:
(1): We provided clarifications around eq. 3 including a new endnote 2. The assumption (stated above eq. 3) is that the filament undergoes a uniform uniaxial extension.
(2): We agree. The argument is now simplified to avoid a new length-scale, and a qualitative physical explanation is added.
(3): We amended both Figures 1 and 2 and their captions to better clarify the flow geometry. The three-dimensional cylindrical coordinates are now explained in the new inset in Figure 2.
(4): The caption of Figure 4 is expanded accordingly (to stress the connection with Figure 3).
(5): A sentence concerning future studies is added at the end of the Summary.